# Psychosocial and Behavioral Effects of the COVID-19 Pandemic on Children and Adolescents with Autism and Their Families: Overview of the Literature and Initial Data from a Multinational Online Survey

**DOI:** 10.3390/healthcare10040714

**Published:** 2022-04-12

**Authors:** Helene Kreysa, Dana Schneider, Andrea Erika Kowallik, Samaneh Sadat Dastgheib, Cem Doğdu, Gabriele Kühn, Jenny Marianne Ruttloff, Stefan R. Schweinberger

**Affiliations:** 1Social Potential in Autism Research Unit & Department of General Psychology and Cognitive Neuroscience, Friedrich Schiller University Jena, 07743 Jena, Germany; andrea.kowallik@uni-jena.de (A.E.K.); samanehsadat.dastgheib@uni-jena.de (S.S.D.); jenny.ruttloff@web.de (J.M.R.); 2Social Potential in Autism Research Unit & Department of Social Psychology, Friedrich Schiller University Jena, 07743 Jena, Germany; msdanaschneider@gmail.com (D.S.); cem.dogdu@uni-jena.de (C.D.); 3DFG Scientific Network “Understanding Others”, SCHN 1481/2-1, 10117 Berlin, Germany; 4Early Support and Counseling Center Jena, Herbert Feuchte Stiftungsverbund, 07743 Jena, Germany; kuehn@stiftungsverbund.de; 5Department of Psychiatry, Jena University Hospital, 07743 Jena, Germany

**Keywords:** COVID-19, autism, children, adolescents, quality of life, well-being, intervention, families

## Abstract

Since COVID-19 has become a pandemic, everyday life has seen dramatic changes affecting individuals, families, and children with and without autism. Among other things, these changes entail more time at home, digital forms of communication, school closures, and reduced support and intervention. Here, we assess the effects of the pandemic on quality of life for school-age autistic and neurotypical children and adolescents. First, we provide a comprehensive review of the current relevant literature. Next, we report original data from a survey conducted in several countries, assessing activities, well-being, and social life in families with autism, and their changes over time. We focus on differences between children with and without autism from within the same families, and on different outcomes for children with high- or low-functioning autism. While individuals with autism scored lower in emotional and social functioning than their neurotypical siblings, both groups of children showed comparable decreases in well-being and increases in anxiety, compared to before the pandemic. By contrast, decreases in adaptability were significantly more pronounced in autistic children and adolescents compared to neurotypical children and adolescents. Overall, although individual families reported some positive effects of pandemic restrictions, our data provide no evidence that these generalize across children and adolescents with autism, or even just to individuals with high-functioning autism. We discuss the increased challenges that need to be addressed to protect children and adolescents’ well-being under pandemic conditions, but also point out potentials in the present situation that could be used towards social participation and success in older children and young adults with autism.

## 1. Introduction

Across the globe, the COVID-19 pandemic has caused a state of crisis, in which local policymakers have faced the ongoing challenge of keeping infection numbers under control, while minimizing negative side effects on mental health and economic prosperity. Correspondingly, behavioral scientists have started to investigate how basic psychological needs, such as maintaining positive relationships, making autonomous decisions, and mastering challenges, have been affected by this crisis [1,2]. Specifically, for children and adolescents with autism [3] and other developmental conditions, many support and intervention programs have either disappeared, or have been restructured to video-based offers that meet the requirements of social distancing.

In this paper, we focus on current knowledge regarding the effects of the pandemic on school-age children with autism and their families in three partially independent sections. First, having closely tracked the rapidly growing number of academic publications over the first 18 months of the pandemic, we share this comprehensive overview of relevant literature with the academic community, although it exceeds the level of detail usually found in the theoretical introduction of a research report. Next, we report data from a pre-registered study investigating changes in anxiety levels, emotional functioning, well-being, and coping of school-age children with clinical autism, compared to neurotypical siblings. This online parent survey was conducted in four languages between July and October 2020 to see how well the themes identified in the literature review reflected the actual experiences of children with autism and their families. Finally, we discuss practical implications and highlight current gaps in research and future perspectives, hopefully inspiring further much-needed research in this field. At the time of writing, the pandemic continues to affect peoples’ lives around the globe, and will continue to do so in the foreseeable future. Thus, our overarching aims are to identify specific difficulties confronting individuals with autism in dealing with pandemic restrictions and to highlight promising strategies for dealing with these difficulties.

In fact, there are several reasons why the pandemic may have affected autistic children and adolescents specifically. Autism spectrum disorder (ASD, hereafter “autism” for short) is a neurodevelopmental disorder characterized by impaired social communication and interaction (e.g., abnormal eye gaze, reduced perspective-taking, stereotyped speech), as well as restricted or repetitive patterns of behavior [4]. The symptoms usually become apparent from early childhood and persist throughout life. Importantly, autism can occur both with and without accompanying impairments of language and intellect. Thus, the exact profile and severity of symptoms in individuals with ASD, as well as their personal strengths and coping abilities, vary widely and can change over time, while the same is true regarding their personal needs for support. The pandemic-related restrictions have undoubtedly played a huge role for families, and especially for those with children with neurodevelopmental, intellectual, or psychiatric disorders. Autistic individuals frequently struggle with social communication and interaction, and some of the changes to daily life have specifically affected these areas (e.g., contact restrictions, physical distancing, video-mediated communication). It thus seemed particularly important to investigate the consequences of these changes for mental well-being and quality of life in this group.

Furthermore, we focused on school-age children and adolescents with autism because many pandemic restrictions have particularly targeted schooling and education. At the same time, the age-range of the individuals reported on in different publications and of the respondents to our own online survey varies quite considerably. This is not surprising, since some consequences of the pandemic have influenced a particular age group, while others can be expected to affect children, adolescents, and adults in a similar way. Note that we will sometimes just use the term “children” in the interest of brevity and readability; however, this should generally be assumed to refer to individuals up to approximately 20 years of age.

Last but not least, due to the world-wide nature of the pandemic, we were particularly interested in recording research findings and survey responses from different countries, cultures, and language backgrounds. This is of interest because the pandemic and the measures implemented to counter it differed between countries, and also because societies vary widely in how they view and support autistic children and their families [5]. Of course, a comprehensive comparison of pandemic effects world-wide goes well beyond the scope of this article; however, we try to maintain a multinational perspective whenever possible.

## 2. Comprehensive Review of the Literature

In the past two years, new scientific papers on the psychosocial and behavioral effects of the COVID-19 pandemic on children with autism and their families have been published at a remarkable pace, though—unsurprisingly—not all of these include new data. Several contributions provide practical advice to parents, caregivers, and therapists of children and adolescents with autism, while considering specific characteristics of the autistic spectrum [6]. Furthermore, while a considerable proportion of the publications are case studies, e.g., [7,8,9], a few large overviews have been presented, e.g., [10].

### 2.1. Search Strategy

To identify those publications most relevant to our research question among the massive number of publications in the wake of the COVID-19 pandemic [11], we used Web of Science as our search engine of choice, and began to search for academic publications in May 2020 using the search terms “((COVID OR Corona) AND (ASD OR autis*))”, for the publication years “2020 OR 2021”. This ongoing review of the literature has continued until now; however, here, we only present papers published up until 21 June 2021. We initially identified 226 papers, of which 84 were excluded because only one of the search terms was relevant (*n* = 80), because they were duplicated (*n* = 1), or because we were not able to access them (*n* = 3). Each of the 142 remaining papers was categorized into one or more of the five topic clusters we identify below, all of which were found to be covered by several papers. Additionally, we distinguished three types of papers: quantitative studies (*n* = 63), qualitative studies including case reports (*n* = 20) and opinion papers, and topical reviews or editorials (*n* = 59). Note that we adopted a rather liberal criterion for classifying a study as “quantitative”, whenever the study methods were described in some detail, even if only descriptive data (e.g., percentages) were reported.

In the following, we begin by describing geographical and temporal aspects of scientific the publications we review. Next, we provide a brief summary of current knowledge in five different topic clusters that were found to be discussed frequently in the literature, and that clearly play an important role for autism under pandemic conditions, including well-being, schooling, leisure activities, comorbidities, and intervention techniques (telehealth and cognitive and behavioral methods).

### 2.2. Regional and Temporal Factors

Around the world, the temporal development of the pandemic and the respective responses by policymakers have varied considerably. Accordingly, it is important to keep in mind that empirical data need to be seen in these spatio-temporal contexts. Pandemic effects on children are likely to differ substantially between a complete lockdown with effective confinement to one’s own indoor living space for weeks (as was the case initially in Wuhan, China, and in Italy in early 2020) versus a lighter version of lockdown in other regions or in phases with moderate infection numbers.

Initial quantitative evidence related to autism quickly became available from China [12], Italy [13], the UK [14], the USA [15], Australia [16], Saudi Arabia [17], Iran [18], or Japan [19], and further qualitative research was reported from the Philippines [20], Zimbabwe [21], or Peru [22]. Topics were diverse. For instance, a preliminary report on more than 3000 families in China [12] focused on caregivers’ well-being and reported that 40–50% reported anxiety, depression, or stress during the first wave of the pandemic. An early paper from Italy [13] reported on an initiative in which parents of autistic children were trained and supported remotely via a telehealth system to continue their applied behavior analysis (ABA) intervention. Despite the fact that the complete lockdown meant that families were effectively on their own all day long over many weeks, this intervention proved very successful and resulted in several positive reports from parents.

Regarding country of origin, our survey reveals substantial numbers of contributions from the USA and the UK, whereas relevant reports from many other countries (including Germany) were slower to emerge (Figure 1 for a global overview of quantitative data). Notably, there is a lack of studies that include longitudinal analyses of pandemic effects at various timepoints or data from more than one country (though see [15,23]). The interested reader is directed to [24], which integrates a number of short comments and perspectives about the situation until June 2020, from editorial board members and global senior leaders of the International Society for Autism Research (INSAR).

### 2.3. Well-Being of Parents, Caregivers, and Individuals with Autism

Difficulty in dealing with change is an important diagnostic criterion for autism spectrum disorders [4]. Consequently, it is not surprising that the rapid changes brought about by the pandemic have prompted numerous authors to address the well-being of families with autistic members. In a theoretical discussion, Bellomo et al. [25] pointed out the risks of delayed diagnoses of ASD and comorbidities, as well as a reduced number and quality of therapy options, triggering concerns about globally poorer health outcomes in the future. An early editorial perspective article by Jefsen and colleagues [10] discussed a survey of more than 60,000 clinical notes on case records from Denmark collected during February and March 2020, as a first step to identify pandemic-related psychopathology in children. The authors hypothesized that children with autism could be particularly vulnerable to distress related to the loss of daily routines. At the same time, they noted that their report could be biased in the sense that it focused on pandemic-related psychopathology only, whereas the number of individuals whose condition might actually have improved was unknown.

Courtenay and Perera [26] and Newbutt et al. [27] focused on the difficulties of children with autism to understand the changes and restrictions in daily activities which could lead to mental stress, anxiety, and behavioral problems. They also pointed out the increased likelihood of exploitation and domestic abuse, as children are no longer protected by their usual communities. Their key message was to make and keep services available.

The favored empirical approaches to gain insight into family well-being were parent surveys and interviews. In self-reports, most parents reported the phase of the first lockdown in Europe as a challenging period [28]. They described themselves as highly stressed [29,30,31,32], more anxious [33,34], and impaired in their emotional well-being [17]. Compared to parents without autistic children, caregivers of autistic children reported higher anxiety levels [35,36]. Similarly, adult siblings serving as caregivers for their siblings with intellectual disability expressed worries about the future, stress, and guilt [37]. Factors including younger age of children and higher severity of autism [31], as well as the amount of externalizing behavior [32], were associated with even greater parental stress. Alyoubi et al. [38] found a correlation between access to appropriate help on the one hand, and less perceived guilt as well as more confidence of parents on the other hand. Positive aspects of the pandemic were also reported, including improved family relationships, quality time together [39,40], and feeling more present and peaceful [41,42]. Mostly, however, parents reported that their affected children showed changes in behavior, poorer emotion regulation, and higher levels of anxiety, especially when routines were distorted [43,44], compared to fewer [35] or less profound changes in respective control groups [14,34]. Changes in behavior included increases in ASD core symptoms, such as stereotypical behaviors [10] and decreased social skills [45], but also in aggression, hypersensitivity, sleep or appetite alterations [46,47,48,49,50], and fewer prosocial behaviors [14]. A meta-analysis comparing children with and without autism in the pandemic revealed worsened behavioral and psychological symptoms, especially in the children with autism [51]. An Italian parent survey based on more than 6000 children [52] found enhanced levels of sleep disorders overall; however, children with autism showed no greater difficulty in falling asleep or staying asleep than other children. Similarly, Cantiani et al. [53] did not find differential effects of pandemic restrictions on the well-being of preschool children with high familial risk for a developmental disorder, compared with typically developing children. At the same time, both groups showed unfavorable changes in their behavioral profiles. Yet again, other findings suggest no change in symptom severity, or in the case of preschoolers with autism, even an increase in adaptive skills, compared to before the pandemic [54].

Self-reporting adults with autism also reported negative effects of the pandemic on their mental well-being, caused, for example, by uncertainty and disturbance of routines [23,55]. Those with higher pre-pandemic levels of depression and anxiety tended to be more severely affected by COVID-related distress [56]. Consequently, an increase in psychiatric hospital admission rates also applied to people with autism during the first lockdown [32]. In contrast, however, one study actually found an improvement in psychological symptoms and a reduction in stress levels as a consequence of the lockdown in adults with autism [20,57].

To summarize, the concerns raised by practitioners and scientists about negative effects of the pandemic are supported by the majority of surveys conducted with individuals with autism and their families. As routines and help services were disrupted, the pandemic placed even more pressure on families, resulting in reduced well-being in children and adolescents with autism and their caregivers [36,40,43,58,59]. This rise in stress and anxiety levels may result in a general increase in mental disorders in the affected families.

### 2.4. Schools, Home Schooling, Home Office, and Work Opportunities

Governmental regulations led to drastic changes in the educational landscape. These changes were often implemented with the needs of the majority of children in mind, while the circumstances of children with special needs were not always adequately addressed [60]. School closures and distance learning became a new global phenomenon, resulting from the requirement to slow down the spread of the virus. During the pandemic, about three-quarters of children with autism initially lost at least one of their educational services, and about half of them received at least one service through tele-education, as suggested by data obtained in April and May 2020 mostly from the USA [15]. Tele-education was rated to be helpful, to some extent, by most parents. Those who received more tele-education rated such services as more helpful. Parents wished for more frequent tele-education, a continuation of the previously provided services, and more in-person services. To meet the need for routines, experts advised affected families to maintain dedicated time slots for schoolwork and to connect regularly with classmates and teachers [6,19]. Stenhoff and colleagues [61] developed detailed guidelines for how teachers and parents can support students’ learning, e.g., by providing a well-structured learning environment and clear instructions. Qualitative research from Zimbabwe and the Philippines found that families adjusted their routines, implemented more diverse learning environments (e.g., educational games), and gave some parents the opportunity to reinforce traditional gender roles during home education [20,21]. Finally, positive findings reported for autistic children were an increase in linguistic abilities [40,48,49] or the general amount of communication [39] during the stay-at-home period.

To summarize, pandemic restrictions around the world resulted in a large proportion of schools closing at least temporarily, often replaced by some level of tele-education. Experiences with this form of schooling were mixed, especially as the special needs and challenges of children and adolescents with autism were not always given sufficient consideration. Many studies particularly highlight the importance of implementing clear routines and a structured time schedule to support distance learning from home.

### 2.5. Leisure Time, Physical Activities, Internet, and Social Media Use

In many countries, recreational activities were restricted as protective measures against the spread of the virus. Correspondingly, there is evidence for reduced physical activity, accompanied by an increase in screen time in adolescents with autism [62]. Since individuals with autism are generally more prone to being overweight and leading a sedentary lifestyle [63,64], the restrictions in mobility [65] as well as online learning environments have likely added to these problems [30]. In consequence, detailed advice was provided to promote home-based physical activity [66,67]. Another line of advice suggested structuring leisure time, e.g., by assigning different rooms for different activities, establishing rules for gaming and using the internet, or creating family playtimes [6,19]. The difficulty of being at home all the time was indeed reported as the primary challenge by parents in the USA [41]. Implemented daily schedules, optimized for the child’s needs, were reported as beneficial [45]. In general, parents had positive opinions about physical activities (e.g., health, social, and psychological benefits); however, barriers such as having to work, security concerns, or insufficient online or tele-education made the implementation difficult.

To mention some examples, a judo program that switched from an in-person to an online live-streamed format was well attended by teenage participants with ASD, and was perceived as helpful in providing structure and physical activity [68]. In a study from Japan, internet use in children and adolescents both with and without autism increased substantially during the pandemic [19]. The increase was even more pronounced in the control group than in the ASD group (+2 h vs. +1.25 h per day), perhaps because the control group had lower pre-pandemic internet or digital media use (median = 2 h per day) than the ASD group (median = 3 h per day). Brondino et al. [69] reported on young adults with severe autism treated at a daycare center where all structures were maintained but adapted to the pandemic situation (e.g., swimming and contact sports were replaced by daily trekking). Importantly, these authors reported no significant worsening of problematic behavior, compared to pre-pandemic times.

To summarize, pandemic restrictions led to considerable increases in screen time for children and adolescents, both due to tele-education and online interventions, but also as a result of the lack of other occupations. As a result, one of the key challenges faced by families in the pandemic was to try to counteract this sedentary life-style, despite reduced opportunities for leisure activities or even restrictions on leaving the house. On a positive note, online exercise courses or the replacement of hygienically problematic physical activities with alternatives seem to have been beneficial in many cases.

### 2.6. Modulating Effects of Level of Functioning, Additional Intellectual Disabilities, and Other Developmental Disorders

Several authors emphasize that pandemic disruptions of daily routines and predictability can be challenging for individuals with autism. Adverse effects could be particularly pronounced for children with intellectual disabilities, who might have difficulty understanding the context of these changes [70]. Conversely, it remains possible that some high-functioning children with autism could even respond positively to the reduction in stressful direct face-to-face social interaction. In many ways, digital social communication and telehealth systems may promote a more predictable and controlled mode of interaction, thus resonating well with autistic individuals’ tendency to “systematize”, i.e., their drive to predict, control or construct systems [71]. This could explain why technology-based interventions may be promising for people with autism (for reviews, see [72,73]). In school contexts, this is illustrated by case reports of two boys with high-functioning autism [74] who enjoyed and even excelled in distance learning. The author speculated that distance learning can eliminate the demands of a “hidden curriculum” of subtle rules for classroom behavior. This could permit children with autism to focus their cognitive and emotional resources on the formal curriculum, improving both performance and life quality.

However, quantitative evidence for whether the level of intellectual functioning affects the extent to which individuals with autism have benefitted from telehealth offers in the context of the pandemic is largely missing. One study indicated that children with intellectual disabilities, but not those with ASD per se, showed increased depression and anxiety levels as a consequence of school-closures [50]. In a large US dataset of 3502 parent surveys conducted in March and April, 2020, White et al. discussed parents’ evaluations of online or telehealth offers [58]. One important finding was that reported benefits increased with the age of the recipients, particularly for special education and mental health services, while they were minimal for pre-school children. Regarding parental well-being, one study reported that a comorbidity of ADHD alongside ASD had only a minimal additional negative effect [28].

To summarize, more conclusive research is still needed concerning how the level of impairment through an individual’s autism and potential intellectual disabilities or comorbidities affect their experience of the pandemic.

### 2.7. Diagnosis and Intervention

A decrease in therapy options and a simultaneous increase in demand for healthcare support—at home or center-based—was most commonly reported by parents during lockdown [28,48,75,76]. Several interventions were presented to support people with autism to make sense of changes and increase compliance and safety. For example, cartoons [77] and social stories helped visualize more abstract facts. One intervention study used modeling and graduated exposure techniques to make autistic children feel comfortable wearing a mask over longer periods of time [78].

#### 2.7.1. Telehealth

Periods of home confinement have rapidly given rise to remote forms of healthcare interventions and trainings [79,80,81] to maintain patient care. At the same time, they have led many affected parents to try existing telehealth offers for the first time [76]. This shift to telehealth offers unique advantages, such as the provision of new insights into children’s daily routines and greater flexibility in scheduling meetings. Disadvantages are the lack of traditional rapport-building activities and visualizations, distractions in the home environment, exacerbated communication difficulties [82], and a lack of physical support and guidance [83]. It is also important to note the limited availability of telehealth technology for low-income families [7,40].

Among other formats, telemedical approaches to monitor infants at risk for ASD [84] have been developed, as well as online assessment tools [85,86] (see also [87] for a comprehensive report on observational tools). Furthermore, complete working models for ASD diagnosis and intervention planning have been established [88,89]. Loman et al. [90] discuss challenges and practical recommendations for remote diagnosis and intervention in autism. Although the gold standard *ADOS-2* cannot be administered validly in full without face-to-face interaction, these authors make useful recommendations for the best practices under conditions of social distancing. In future, a coherent implementation of such new workflows, including telehealth, could lead to better use of professional resources, improved access especially in rural areas, reduced waiting times, and higher care quality, compared to pre-pandemic times [91,92,93]. Among adults who were currently undergoing a diagnostic process for either ASD or ADHD, the telemedical approach was rated as useful and effective in this respect, even though roughly half of the participants would have preferred face-to-face contact [94]. Nevertheless, even best-practice telehealth did not always achieve the goal of improvement and left parents and care teams with unresolved questions on how to proceed [95]. During the first months of global lockdown, caregivers of persons with genetic neurodevelopmental disorders (40% ASD) did not experience the available telehealth services as helpful [96]. Case studies suggest that difficult-to-treat cases were particularly compromised by the lack of on-site treatment during the lockdown periods [9].

To provide some examples, Beaumont et al. [16] adapted an existing computer game-based social skills program for autistic children to a caregiver-supported version. This adaptation improved social skills and problem behavior in the treatment group compared to a no-training control group. Similarly, a remote audio coaching for young adults with intellectual disabilities taught small talk skills effectively [97]. A further adaptation for telehealth was the *Cool versus Not Cool* intervention, which teaches specific social behaviors to children with autism and was able to achieve specific pre-determined goals even via telehealth, in 4–8 sessions, albeit for a very small sample [98]. Creative play treatments were also adapted successfully to a one-on-one online format that seemed to increase motor, balance, and imitation skills [99]. Another approach focused on empowering parents of young autistic children by providing an entirely online 12-week program focused on emotion regulation and challenging behavior management [100]. In addition to providing individualized coaching for dealing with specific challenges, a positive side benefit was the emergence of support networks among parents. In another telehealth study, professionals instructed parents about training daily living skills with their children. Within their limited sample, the majority of activities were taught successfully, suggesting that parent-implemented interventions can be promising [101]. Finally, child and adolescent psychiatry practitioners in the USA were also targeted by training programs themselves, e.g., an ASD-specific training on sexual health which consisted of video clips and video-conference meetings. This training significantly increased practitioners’ skills, knowledge, and positive attitudes, while achieving a countrywide outreach [102].

#### 2.7.2. Cognitive and Behavioral Methods

Especially among providers of applied behavior analysis (ABA), there has been an ethical discussion about the decision-making process for in-person treatments under pandemic conditions. Cox et al. [103] argued that the risk of continued in-person treatment outweighed possible benefits for many clients and that temporal suspension of services or telehealth might often be the preferred option. Others have recommended implementing adequate safety procedures [104] in order to maintain 1:1 treatment as far as possible. This could even entail in-person treatments if necessary [105] and maintaining a physical setting [106]. After reviewing the limited data available, Schlietz and Wacker [107] concluded that telehealth can in fact be equally effective to in-person ABA.

During the first lockdown, most in-person therapies were at least partially suspended. However, a study of ABA in young children aged 15–30 months showed that the training effects of therapy were maintained over a 3-month lockdown period. Behavioral problems became more apparent again when training resumed [108]. Yi and Dixon [109] provide details on implementations of their telehealth model of ABA, Belisle et al. [110] provide detailed technical instructions for converting tasks into an online format, and Corona et al. [111] added the clients’ perspectives to the decision-making process by providing a caregiver interview to assess demand for changes in frequency or setting. As previously mentioned, Espinosa et al. [13] developed a very intense ABA-based telehealth intervention program for parents, which involved teaching them to implement daily structures, as well as choosing appropriate activities and ABA-specific interventions (e.g., discrete trial teaching). Reassuringly, Pollard et al. [112] showed that if the number of training hours was maintained and the form of instruction (e.g., parent-instructed, telehealth parent-assisted, or telehealth directly with the practitioner) was chosen according to the child’s abilities, most children continued to benefit from ABA training.

Another form of pandemic-adapted training is based on acceptance and commitment therapy (ACT). For example, the detailed guidelines by Szabo et al. [113] promote parents’ use of schedules and routines, family values, norms and positive reinforcement by suggesting specific exercises that align with the values of individual families. The *5 Cs* method (i.e., self-control, compassion, collaboration, consistency, celebration) for supporting families with alternative-learning children was also discussed in light of the current pandemic [114]. Tarbox et al. [115] adapted well-known ACT exercises to the pandemic situation, implemented into an ABA-based therapy framework to help children cope with the changes. A combined approach used a behavioral plan combined with a mindfulness-based intervention to reduce self-injury behavior in autistic adolescents has shown initial promising results [116].

Kalvin et al. [82] suggested modifications of a telehealth-based cognitive behavioral training for anxiety in children with autism. They detected specific changes in anxiety, depending on the type of anxiety (e.g., decreases in social anxiety, increases in fears of contamination or family safety) that made re-evaluations of goals and anxiety hierarchies, and subsequently new exposure exercises, necessary. They argue for enabling parents to provide developmentally appropriate information about COVID-19 to their children, while also setting realistic expectations.

To summarize, a large proportion of the published literature to date has focused on adapting existing diagnostic tools and interventions to an online format for use under lockdown and social distancing conditions. Several of these programs show promising effects, though limited access for many families pose a serious challenge. Another important issue is the ethical decision of assessing the relative risks of maintaining on-site services vs. remote and technically mediated contact.

### 2.8. Further Topics of Interest

In the studies we reviewed, we also identified several themes that were covered only occasionally and thus did not fit into any of our topic clusters described above. Some will be highlighted in the following, particularly where they cross the borders of social behavioral and medical sciences. For instance, in an editorial letter, Baghdadli et al. [117] pointed out that autistic people with more severe intellectual disabilities may be particularly vulnerable, both due to their living situations, but also because they can have immune systems with increased levels of inflammatory cytokines, putting them at increased risk for severe forms of COVID-19. Another important issue raised by Türkoğlu et al. [118] is sleep problems of children with autism during the COVID-19 pandemic, their relationship to symptom severity, and perspectives for reducing sleep problems to mitigate negative effects.

Special needs should be considered for people with autism who are hospitalized with COVID-19. Two papers from the same institute in Paris, France [119,120] report initial experiences and quantitative data from a COVID unit for autistic patients only, created in an interdisciplinary collaboration between child and adolescent psychiatry and specialists for infectious diseases. This unit made it possible to respond to specific psychological and medical characteristics of patients with autism, such as a higher prevalence of gastrointestinal or immunological disorders. Amongst other hospital measures, when ventilation is needed for an individual with autism, non-invasive methods should be systematically considered to reduce anxiety and arousal [121]. Outside the hospital environment, better routines need to be developed and evaluated for home-based testing for COVID-19 infections in individuals with autism, in a way that considers sensory concerns and compliance [70]. Despite promising local initiatives, a review of policies across 15 European countries suggested that the COVID-19 pandemic has further accentuated healthcare inequalities for individuals with autism, and that these policies need revision to counteract disproportionate risks to physical health and quality of life [122].

Although this article focuses on children and adolescents, changes relating to physical distancing affect both children and adults, in schools and in the employment sector. Regarding prospects for work opportunities in a difficult economy, a study from the UK assessed 78 individuals with mental handicaps, who reported that the most helpful provision for being able to work efficiently was a safe and quiet place to go when overwhelmed [123]. Data for that study were collected in summer 2019, but the results were discussed in the context of the pandemic. The paper, published in an outlet in business and economy, gives an impressive account of how societies currently waste potential to empower adults with autism, and how such empowerment can result in both integration and economic and financial success. As telework often seems to provide a beneficial context for good performance, the pandemic could provide a unique window of opportunity in several fields. Ultimately, this will not only be important for children and adolescents, but also for employers and many adults with autism. As a possible qualification, a small longitudinal study in working adults with autism suggested a substantial decrease in mental health for those who lost their jobs during the pandemic, as well as a marginal decrease in mental health for those who transferred to working from home, compared to those who continued physically attending work [124]. More studies with larger and more representative samples will be necessary to further clarify implications of the pandemic for working adults with autism.

### 2.9. Summary of Reviewed Work

In our comprehensive review of the literature published within the first 18 months of the COVID-19 pandemic, we surveyed substantial initial data collected across the globe. A large proportion of this work addressed well-being and intervention; however, data are also beginning to permit regional–temporal comparisons and provide information on schooling or leisure time changes. In sum, children and adolescents with autism tended to show less adequate emotion regulation and higher stress levels than neurotypical children, particularly when daily routines were disrupted. An increase in ASD core symptoms (i.e., stereotypical behaviors and decreased social skills), as well as increases in aggression, hypersensitivity, sleep, and appetite alterations, were reported. Additionally, the well-being of caregivers was affected by anxiety, depression, and stress, as established routines, help services, and schooling and work opportunities were no longer available, or only in limited forms. To meet the special routines required for individuals with ASD, various studies made clear that maintaining dedicated time slots for school/work and leisure time, despite the altered environment, could protect against increases in ASD symptoms and general mental health deterioration.

Many contributions related to diagnosis and intervention pointed out that the COVID-19 pandemic entails a high risk of leaving individuals with ASD behind. One encouraging development that may in part counter this risk is an impressive surge of telehealth implementations which—to mention some examples—explain new hygiene practices using cartoons and social stories, provide tele-diagnostic instruments even for very young children, and implement telehealth set-ups for established cognitive and behavioral therapies and social skills programs. Importantly, the benefits from online or telehealth offers appear to increase with the recipients’ age, particularly for special education and mental health services, and their usefulness is severely limited for pre-school children. Positive aspects of the pandemic were sparsely reported, but included improved family relationships and gains in linguistic abilities and communication. Overall, the new situation has brought big challenges and a high risk of leaving a vulnerable population behind, while at the same time giving rise to a use of technology that might provide valuable additional resources in the post-pandemic world.

## 3. Online Survey

In order to gain insight into how well the themes identified in the literature review reflected the actual experience of children with autism and their families, we conducted an online parent survey focused on assessing activities, well-being, and social life in families with autism, as well as changes at various stages of the pandemic. We aimed to compare pandemic effects on children with and without autism, ideally from within the same families. In particular, we were struck by the observation that while social interactions during the COVID-19 crisis have become highly constrained, they have also tended to be more predictable under normal circumstances. A working hypothesis, confirmed in discussions with practitioners from the local Early Support and Counseling Center, was that some children with autism might actually benefit from the novel situation. In this case, parent-rated changes in anxiety levels, emotional functioning, social functioning, well-being, and coping of school-age children with clinical autism spectrum diagnoses would actually be more favorable than for their neurotypical siblings. This hypothesis was preregistered on 16 June 2020 (https://aspredicted.org/us3py.pdf, accessed on 5 April 2022). The research was approved by the Ethics Committee of the Ethical Commission of the Faculty of Social and Behavioral Sciences of the Friedrich Schiller University Jena (FSV 20/025).

### 3.1. Materials

Four parallel versions of an online survey were generated using SoSci Survey [125] and made available to users via www.soscisurvey.de (accessed on 5 April 2022). The four versions differed only in the survey language: English, German, Turkish, and Persian (Farsi), each translated by a native/fluent speaker of the respective language. The English version is provided in the Appendix A, and all four language versions are available at https://osf.io/sm3ze/?view_only=a9b5df135dc64e82bff0512c85fcd707 (accessed on 5 April 2022).

Among other information, the survey assessed demographic information about the responding parent, the children with and without autism, the rest of the family, and the child’s autism diagnosis (using the AQ-10 child version) [126]. We also asked about living conditions, local pandemic restrictions, and the parent’s personal views on the pandemic. For both the child with autism and the neurotypical control child, parents indicated how much time per day they spent on various activities (e.g., computer games, online learning, sleeping, and playing outside) before the pandemic, at the peak of the first wave, and at the time of answering. In addition, they reported on how well children complied with hygiene rules and filled in the PedsQL™ General Well-Being Scale-Pediatric Quality of Life Inventory™ General Well-Being Scale (PedsQL, Mapi Research Trust) [127] in English, German, Turkish, or Farsi for Iran (contact information and permission to use for all four language versions: Mapi Research Trust, Lyon, France, https://eprovide.mapi-trust.org/instruments/pediatric-quality-of-life-inventory, accessed on 5 April 2022). This is a validated parent-report scale, in which various dimensions of health and well-being are assessed based on 5-point scales assessing each variable over the course of the previous month. Finally, parents were given the opportunity to comment on specific problems, stressors, the provision of support, and the situation in general.

### 3.2. Participants

An initial power analysis using G*Power 3.1 [128] (https://www.psychologie.hhu.de/arbeitsgruppen/allgemeine-psychologie-und-arbeitspsychologie/gpower, accessed on 5 April 2022) suggested that a minimum sample size of 53 per group (autistic vs. neurotypical) would be required to identify a medium-size effect of *d* = 0.5 for a comparison between two groups (nonparametric Wilcoxon–Mann–Whitney *t*-test, one-tailed, alpha set to 0.05, power of 0.80).

Parents were recruited to participate through professional networks, word of mouth, adverts in research and support institutions, and postings in social media. Data were collected in summer 2020, after the first wave of infections in most countries. At this point, infection numbers were relatively low, but respondents presumably still had a fairly fresh representation of their experiences during the first wave. The survey was available online from 16 July to 31 October, with the last dataset entered on 22 September 2020. We stopped data collection when infection rates began to increase significantly, suggesting that the next infection wave was on the way (i.e., 22 September, 2020, 7-day incidence per 100,000 inhabitants: Germany—1771; Iran—3120; Turkey—1672; and UK—4189, according to interactive dashboard data from John Hopkins University, see https://systems.jhu.edu/tracking-covid-19/, accessed on 5 April 2022). During this time, 70 surveys were completed by parents of school-age children and adolescents with a clinical diagnosis of ASD from seven countries (German version, *n* = 37: Germany and Austria; Persian version, *n* = 15: Iran; Turkish version, *n* = 13: Turkey; and English version, *n* = 5: Australia, USA, and UK).

A closer look at the data revealed that only 57 of the children and adolescents had a clear and confirmed clinical diagnosis of ASD. These children (14 female, 25%; 40 male, 70%, 3 undisclosed/diverse) varied considerably, both in age (5–20 years, *M* = 11.09 years, *SD* = 3.97) and in level of required support (3—very little, 26—some, 21—substantial, and 7—very substantial). For comparison purposes, parents were also asked to provide information on one neurotypical child that they were in frequent contact with. This provided data for 53 neurotypical children (23 female, 43%; 29 male, 55%; 1 undisclosed/diverse; age 0–25 years, *M* = 10.0, *SD* = 4.87).

However, not all parent reports contained full data for both one child with confirmed autism and one clearly neurotypical comparison child. Thus, comparisons within families are based on only the 43 cases that reported both on one child with a confirmed autism diagnosis (AQ-10: *M* = 7.65, *SD* = 1.91; 12 female, 28%; 30 male, 70%; 1 undisclosed/diverse, 2%; age: *M* = 10.78, *SD* = 4.07) and on one control child that was indicated as uncontroversially neurotypical (AQ-10: *M* = 1.67, *SD* = 1.84; 20 female, 48%; age: *M* = 10.24, *SD* = 5.16). A *t*-test confirmed that this novel form of familial reporting actually achieved a rather good age match between the two groups (*t*(40) = 0.862, *p* = 0.394; 2 out of 43 pairs not included in this analysis due to missing age data). These 43 families were living in Germany (*n* = 21), Iran (*n* = 9), Turkey (*n* = 8), and Austria (*n* = 5).

### 3.3. Data Analysis

In addition to the analysis based on sibling data from the same family (*n* = 43), we also ran some analyses on the entire dataset of children and adolescents with a clinical diagnosis of ASD (*n* = 57 individuals). This made it possible to collapse children and adolescents with autism into a (comparatively) high-functioning subgroup (requiring very little or some support; *n* = 29; AQ-10: *M* = 7.21, *SD* = 2.21) and a low-functioning subgroup (requiring substantial or very substantial support; *n* = 28; AQ-10: *M* = 8.11, *SD* = 1.47). Please note that in all analyses that compare autistic and neurotypical children, we obtained similar results when using the entire dataset (thus having slightly larger samples at the expense of losing family matching).

As already mentioned, we stopped data collection with 70 surveys in total. For this reason, some of the analyses presented in the following may have low statistical power and should be interpreted with due caution. Initially, we also planned to compare data across several countries and four languages to assess similarities and differences across different world regions. However, our final sample was too small to permit such comparisons with sufficient statistical power. Thus, we integrated the data across the four languages in the present report.

### 3.4. Results

We compared siblings with and without autism from the same families regarding scores on the PedsQL™ dimensions *emotional functioning* and *social functioning*. These scores range from 0 to 100, with higher scores representing higher levels of functioning. Children with autism showed lower levels of both emotional functioning (*M* = 49.3, *SD* = 19.6, 10–95, *n* = 43) and social functioning (*M* = 34.5, *SD* = 21.7, range 0–85) than their siblings without autism (emotional: *M* = 65.0, *SD* = 17.0, range 25–100, *n* = 43; *t*(42) = −4.702, *p* < 0.001; social: *M* = 76.5, *SD* = 23.13, range 10–100; *t*(42) = −9.638, *p* < 0.001). Similarly, considering all children with autism, the high-functioning children (*n* = 29) showed higher levels of emotional (*M* = 57.24, *SD* = 19.67; range 20–100) and social functioning (*M* = 42.76, *SD* = 23.78, range 10–100) than low-functioning children (*n* = 28, emotional: *M* = 45.89, *SD* = 19.1, range 10–100; *t*(55) = 2.209, *p* = 0.031; social: *M* = 29.11, *SD* = 20.82, range 0–85; *t*(54.5) = 2.308, *p* = 0.025). The range of responses was substantial in all groups of children.

Siblings with and without autism did not differ significantly in their reported response to, and compliance with, hygiene recommendations regarding physical distancing, hand washing, and using disinfectant (all *p*s > 0.3), assessed on a 3-point scale from negative (−1) to positive (1). However, children with autism responded less positively to wearing a face mask (*M* = 0.0, *SD* = 0.9) than their neurotypical siblings (*M* = 0.35, *SD* = 0.77; *t*(39) = −2.303, *p* = 0.027, *n* = 43). Considering all children with autism, low-functioning children were reported to respond negatively to face masks (*M* = −0.21, *SD* = 0.92, *n* = 28), while the response of high-functioning children was more positive (*M* = 0.21, *SD* = 0.86, *n* = 29; *t*(55) = 1.788, *p* = 0.079). Interestingly, all other means for hygiene behaviors were positive in all groups of children, suggesting that, overall, most children complied with hygiene recommendations most of the time.

At the time of the parents’ reports, both autistic and neurotypical children showed a decrease in general well-being and social behavior, as well as increased overall anxiety, compared to the time before the pandemic. These measures were assessed using a 5-point scale from much increased (2) to much decreased (−2). An overview is provided in Table 1, which first reveals changes for autistic vs. neurotypical children from within the same families, and second for high- and low-functioning children with autism. Children with autism showed a much larger decrease (*M* = −0.74, *SD* = 1.18) than their neurotypical siblings (*M* = −0.19, *SD* = 0.91; *t*(42) = −2.439, *p* = 0.019; *n* = 43) in adaptation abilities. This pattern was numerically similar in the comparison within autistic children, potentially indicating greater declines in adaptation abilities for low-functioning children than high-functioning children.

Finally, as an additional analysis, we asked about changes compared to the time before the pandemic at the level of entire families with at least one child with autism (*n* = 57). This measure, which was constructed to score from −2 to 2, as above, revealed both negative and positive aspects of the pandemic. On the one hand, the general stress level (*M* = 0.82, *SD* = 1.23) and frequency of family conflicts (*M* = 0.61, *SD* = 1.0) were both reported to have increased considerably (simple *t*-tests against zero: *t*(56) = 5.076, *p* < 0.001, and *t*(56) = 4.656, *p* < 0.001, respectively). On the other hand, marked increases were also reported regarding the amount of time spent together (*M* = 0.93, *SD* = 1.12; *t*(56) = 6.292, *p* < 0.001) and emotional closeness (*M* = 0.54, *SD* = 1.04; *t*(56) = 3.962, *p* < 0.001). Again, parents used the entire scale from −2 to +2 to report on these aspects, suggesting a wide variability in how different families experienced and/or responded to the changes.

### 3.5. Summary and Discussion of the Online Survey Results

The online parent survey, alongside the standardized PedsQL™ dimensions for emotional functioning and social functioning, confirmed that children with autism scored considerably lower than control children who shared the same family environment and situation. On change-based measures of social and behavioral aspects (i.e., general well-being, coping/adaptation abilities, overall anxiety, emotional reactions, and social behaviors) before and at the peak of the pandemic, the extent of negative change was comparable for children with autism and neurotypical children, except for coping/adaptation abilities. The latter was significantly stronger for children with autism compared with neurotypical children. Our working hypothesis of better outcomes for individuals with autism was not confirmed on any measure. Interestingly however, all measures showed a wide spread of responses, indicating that some individuals and/or families struggled hugely under pandemic restrictions, while others coped quite well. Along these lines, some positive findings show that most children responded comparatively well to recommended hygiene behavior, with the exception that autistic children were more hesitant about wearing face masks. Furthermore, and as reported in some papers from our literature review, families spent more time together and experienced greater emotional closeness within the family.

## 4. General Discussion

An efficient response to the challenges of the COVID-19 crisis requires a massive effort from the social and behavioral sciences [2] to reveal more information on potentially devastating effects of the global COVID-19 pandemic on vulnerable individuals. In the current paper, we reviewed publications from the first 18 months of the pandemic containing qualitative and quantitative data on individuals and families affected by autism. Furthermore, we provided results from a parent survey of school-age children and adolescents with autism, conducted from July to October 2020. In this final section, we bring together these different perspectives, discuss new insights and limitations, and integrate these with further literature which has been published since the end of our literature search. We conclude by suggesting that social interaction is a key facet that must be borne in mind in order to alleviate the negative effects of pandemic restrictions on individuals with autism.

Overall, both the main bulk of the literature and the responses to the survey suggest that the pandemic has had many dire consequences for autistic children and their families around the world. In particular, the breakdown of established support networks severely exacerbated the situation in many circumstances and placed a huge burden on the immediate family, who were often the only remaining care-givers. Another very worrying implication is a potential delay in the diagnosis and onset of intervention programs.

At a local level, we can confirm the following. Here in Jena, Germany, the Early Support and Counseling Center was forced by local authorities to stop offering physical special needs and support services in kindergartens, schools, and homes of affected families. The change was sudden, and many parents of children with autism reported that it entailed considerable stress and anxiety due to the loss of daily routines and insights from a third party. It was reported that the younger the affected individuals, the more they struggled with the COVID-19-induced changes. Interestingly, family relationships were generally described as positively different, as more leisure time was spent together than at other times. Similarly, the Early Support and Counseling Center Jena reported that many adults with autism experienced the lockdown as less problematic and sometimes even as relaxing, due to the reduction in face-to-face social contacts. Many clients also reported positive effects on their family life, which they attributed to more time than before the pandemic spent on activities such as hiking, play, or music—again, an observation that was corroborated by our online survey.

From our comprehensive literature review, it became clear that more cumulative research is needed to quantify pandemic effects on individuals with autism requiring various levels of support, as well as to further develop, optimize, and evaluate support offers for schooling and telehealth interventions. Unsurprisingly, there are still critical gaps in knowledge and a global effort is required to integrate data from across the world and from different pandemic phases.

Regarding our parent survey of school-age children and adolescents with and without a clinical diagnosis of autism, we fully acknowledge the limitations of our analyses due to the comparatively small number of respondents and their diverse geographical, cultural, and economic backgrounds. In retrospect, the survey was very long for busy parents to fill out online at a challenging time. It is interesting to speculate about potential response biases here. On the one hand, parents of severely affected children might be particularly likely to fill out the survey in order to vent their frustration; on the other hand, this group is likely to have especially little time to spare for voluntary add-ons such as a survey. Another issue potentially adding to the low response rate is the fact that data collection fell within the summer break in many countries, when daily routines are often suspended even in non-pandemic times. Finally, it goes without saying that the sampled populations in the German-speaking countries, Iran, and Turkey cannot in any way represent a global view. Among many other aspects, they differ on how autism is treated at a societal level [5], as well as with regard to legislative and administrative responses to the pandemic. Nonetheless, we believe that our study provides some useful insights to motivate future research. In particular, we can recommend sampling autistic and non-autistic individuals from the same family or care environment whenever this is feasible, in order to make conclusions more autism-specific and less dependent on societal and cultural influences. Further, change-based measures comparing effects between more relaxed and more constrained pandemic phases (e.g., throughout vs. at the peak of a wave; summer vs. winter; soft vs. hard lockdown measures) also seem important to gain better control of situational and societal differences. On a critical note, third-party reports from parents and primary caregivers of affected individuals with autism may be biased more than, for instance, reports from support services, teachers, therapists, or medical staff, and of course than individuals’ self-reports. At the same time, parent and primary caregiver reports provide a valuable perspective that more objective reports can never provide, especially if the latter pose a social norm that individuals with autism try to adhere to, potentially masking some of their autism characteristics [129].

To conclude, we would like to mention a few positive findings. Although the numbers are hard to interpret, it seems that most children responded comparatively well to recommended hygiene behavior, with the exception that autistic children were hesitant about wearing face masks. When integrated with other findings [78,130,131], we find it encouraging that children with autism can adhere to these recommendations when conveyed appropriately. Consistent with other findings, e.g., [38], the present data indicate that due to either advice or legislation to stay at home, families with children with autism spent more time together, and experienced greater emotional closeness within the family than before the pandemic. Without a doubt, this could contribute to parents’ and children’s ability to deal with the challenges posed by the COVID-19 pandemic.

### Outlook: The Importance of Social Interaction in the Face of the Pandemic

Finally, we would like to reflect briefly from the perspective of social neuroscience on the challenges and changes the pandemic imposes on our societies in general, and on the impact these changes can have on autistic and neurotypical people in the longer term. When considering the psychological [1] and neurobiological effects of social isolation, it is hardly surprising that the degree of isolation that many people experienced during the COVID-19 pandemic poses a threat to both mental and bodily health. In a recent review in response to the pandemic, Bzdok and Dunbar [132] summarize the importance of social bonds for human health and illustrate why some forms of digital communication cannot effectively compensate for real-time face-to-face interaction or are experienced as cognitively and emotionally exhausting [133]. When communicating, our brains are tuned to process multiple simultaneous signals from different senses. For example, prosody and eye contact help to steer conversations and joint action. This requires ongoing processing of dynamic representations of the face, voice, bodily motion, and other activities from an interaction partner [134]. Recent research suggests that social interaction is a critical driving factor that shapes individual brains through lifetime [135] and has shaped the human brain during evolution [136]. Bearing this importance of social interaction in mind, Bzdok and Dunbar [132] consider effective means to mitigate the costs of isolation, such as supporting friendships, mental training towards empathy, taking perspective of others’ mental states, or joint singing.

Although joint singing typically happens in choirs and singing groups, the internet has enabled musicians to perform together, despite requirements of physical distancing via imitation and remix. Of note is the social-media success of Scots’ postman Nathan Evans, who in December 2020 became famous after posting the 19th-century shanty “The Wellerman”. This example, and the overwhelmingly positive response of people suffering from social isolation, could point to a therapeutic form of social interaction via digital media that is not (much) compromised by their technological limitations. In this instance, it may help that familiar music is temporally predictable, making it easy to sing or play along, while eye contact is of little or no relevance. While video-based interaction unarguably suffers from some technological limitations, physical interaction under pandemic conditions is compromised by numerous other obstacles, a prominent one being face masks, which degrade face perception and emotion recognition. In fact, negative effects of face masks on emotion recognition appear to be disproportionately severe in pre-school children, compared to older children or adults [130]. Accordingly, despite their critical importance for slowing the spread of the virus, face masks could effectively contribute to enhancing problems of young children with ASD to recognize facial emotions [137].

Undeniably, people with autism suffer from social isolation, as do neurotypical individuals [23]. Although certain behavioral signs in autism, such as low levels of eye contact, have long been taken as an indicator for a lack of social interest or motivation [138], this interpretation is likely incorrect. Instead, autistic people are more likely to express their social motivation in unconventional manners [139,140]. Mitchell et al. emphasize how being misperceived, against the background of social conventions, creates risks for the mental health and well-being of people with autism. The “double empathy problem” describes how individuals with autism find fitting into society difficult, not only because they misunderstand others, but also because they are misunderstood by others who reflect a bidirectional failure of empathy [140]. In some sense, the COVID-19 pandemic has submitted humankind to a global experiment, in which established social conventions (e.g., handshaking, keeping distance according to one’s peripersonal comfort zone, preferring face-to-face communication) have been eroded or are changing dramatically, creating ruptures in social interaction for everyone. While we need to take the associated risks seriously, the current situation of changing social conventions may also contain the potential to promote a better and more humane understanding between autistic and neurotypical people.

## Figures and Tables

**Figure 1 healthcare-10-00714-f001:**
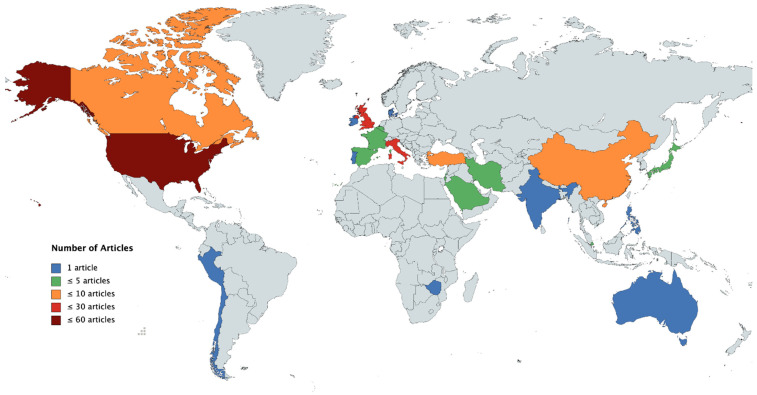
Countries of origin of the 142 papers included in our literature review, color-coded by the number of contributions on a World Map (https://mapchart.net/world.html, free online tool, accessed on 5 April 2022). In detail, contributions came from 23 countries, with information per country on the total number of papers (in brackets separate counts for quantitative studies/qualitative studies/topical reviews, editorials, or opinion papers): Australia (*N* = 1: 1/0/0), Belgium (*N* = 2: 1/1/0), Canada (*N* = 7: 2/0/5), Chile (*N* = 1: 1/0/0), China (*N* = 7: 4/2/1), Denmark (*N* = 1: 0/0/1), France (*N* = 5: 2/0/3), India (*N* = 1: 0/0/1), Iran (*N* = 3: 1/0/2), Ireland (*N* = 1: 0/1/0), Israel (*N* = 2: 1/0/1), Italy (*N* = 17: 9/2/5), Japan (*N* = 4: 3/0/1), the Philippines (*N* = 1: 0/1/0), Portugal (*N* = 1: 1/0/0), Saudi Arabia (*N* = 3: 3/0/0), Singapore (*N* = 2: 0/0/2), Spain (*N* = 3: 3/0/0), Turkey (*N* = 9: 4/3/2), UK (*N* = 19: 6/5/8), USA (*N* = 53: 21/4/27), and Zimbabwe (*N* = 1: 0/1/0). Note: The map data are for 145 papers, including 3 further papers (one from Peru, one from the USA, and one from Italy) which were excluded from the review despite being topically relevant, either because they were written in a different language, or because we had no access to the full text.

**Table 1 healthcare-10-00714-t001:** Overview of mean changes in quality of life and functional abilities, compared to the time before the pandemic (SD in brackets).

Comparison	Sub-Group of Children	General Well-Being	Adaptation Abilities	Overall Anxiety Levels	Emotional Reactions	Social Behavior
Within-family reports (*n* = 43)	Autistic sibling	−0.40 (1.12)	−0.74 (1.18)	+0.44 (0.91)	−0.09 (1.23)	−0.58 (1.22)
	Neurotypical sibling	−0.44 (0.93)	−0.19 (0.91)	+0.47 (0.77)	+0.05 (1.0)	−0.26 (1.11)
Paired *t*-tests		*t*(42) = 0.323, *p* = 0.743	*t*(42) = −2.439, *p* = 0.019	*t*(42) = −0.147, *p* = 0.884	*t*(42) = −0.650, *p* = 0.519	*t*(42) = −1.207, *p* = 0.234
All autistic children with confirmed diagnosis	Low-functioning (*n* = 28)	−0.25 (1.14)	−0.93 (0.98)	+0.43 (1.0)	−0.14 (1.24)	−0.68 (1.28)
	High-functioning (*n* = 29)	−0.34 (1.08)	−0.59 (1.30)	+0.59 (0.87)	+0.10 (1.11)	−0.52 (1.15)
Independent *t*-tests		*t*(55) = −0.322, *p* = 0.748	*t*(55) = 1.128, *p* = 0.265	*t*(55) = 0.636, *p* = 0.528	*t*(55) = 0.789, *p* = 0.434	*t*(55) = 0.50, *p* = 0.619
Total (*n* = 57)		−0.30 (1.10)	−0.75 (1.15)	+0.51 (0.93)	−0.02 (1.17)	−0.60 (1.21)

Note. The 5-point scale ranged from −2 to 2, so negative numbers indicate a decrease in the relevant function, and positive numbers indicate an increase. Paired *t*-tests comparing children with ASD to neurotypical control children were significant regarding adaptation abilities only. Note that degrees of freedom differ since comparisons between autistic and neurotypical children use paired *t*-tests for siblings within families (*n* = 43), while comparisons between high- and low-functioning autistic children are based on independent *t*-tests for all children with autism reported in the survey (*n* = 57).

## Data Availability

The datasets analyzed for this study can be found on OSF: https://mfr.de-1.osf.io/render?url=https://osf.io/3cmg7/?direct%26mode=render%26action=download%26mode=render (accessed on 5 April 2022).

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
