# Peer review of "Psychosocial and Behavioral Effects of the COVID-19 Pandemic on Children and Adolescents with Autism and Their Families: Overview of the Literature and Initial Data from a Multinational Online Survey"

_healthcare, 2022, doi:10.3390/healthcare10040714_

Round 1
Reviewer 1 Report
First of all, I would like to thank you for the opportunity to read this very interesting manuscript. I enjoyed reading it very much, as it deals with a crucial topic within the field of special education and the necessary attention that this group should have from different fields, including the clinical one.
When I started reading the article I thought it was a review study. Later, I realized that it also had an empirical part. The first part of the study corresponds to a review of the literature, where a global view of the topic to be studied is provided. In this regard, the authors have not been able to contextualize the problem in question. Although the theoretical framework is well supported, it would have been essential to elaborate a characterization of this group in order to understand how the pandemic has affected them so negatively.
In this theoretical framework the authors provide the flowchart typical of statements such as PRISMA. The inclusion and exclusion criteria are not clearly stated. The authors refer to Figure 1 to intuit such criteria. Once the first part of the study is finished, the authors include the methodology section, but this confuses the reader as it seems to be a different work.
They have administered a questionnaire to parents to assess how the pandemic has affected their autistic children, but there is a lack of data on how the contact with those parents and the process developed.
The results, discussion and conclusions are consistent.
The main problem of this work is the absence of a common thread that allows the understanding of the work. I recommend the authors to suppress the flowchart and change the structure of the paper.
At the content level, except for the method section that should be better explained, the article is adequate.
Author Response
First, we would like to thank both reviewers sincerely for their quick but very supportive, to-the-point and helpful reviews. We feel that the changes initiated in response to their comments have really improved the manuscript substantially by increasing readability and integrating the separate parts of the article in to a much more coherent whole.
Point 1: When I started reading the article I thought it was a review study. Later, I realized that it also had an empirical part. The first part of the study corresponds to a review of the literature, where a global view of the topic to be studied is provided. In this regard, the authors have not been able to contextualize the problem in question. Although the theoretical framework is well supported, it would have been essential to elaborate a characterization of this group in order to understand how the pandemic has affected them so negatively.
Response 1: We have used the first section (i.e. before Section 1) to clarify in more detail that the structure of the paper, and in particularly the literature review, is slightly unusual. In addition, we have added a paragraph here describing Autism Spectrum Disorder in more detail and elaborating on how the pandemic might have affected this group of people specifically:
“In fact, there are several reasons why the pandemic may have affected autistic children and adolescents specifically. Autism Spectrum Disorder (ASD, hereafter “autism” for short) is a neurodevelopmental disorder characterized by impaired social communication and interaction (e.g., abnormal eye gaze, reduced perspective-taking, stereotyped speech), and restricted, repetitive patterns of behavior [4]. The symptoms usually become apparent from early childhood and persist throughout life. Importantly, autism can occur both with and without accompanying impairments of language and intellect. Thus, the exact profile and severity of symptoms in individuals with ASD as well as their personal strengths and coping capabilities vary widely and can change over time, while the same is true regarding their personal needs for support. The pandemic-related restrictions have undoubtedly played a huge role for families, and especially for those with children with neurodevelopmental, intellectual or psychiatric disorders. Autistic individuals frequently struggle with social communication and interaction, and some of the changes to daily life have specifically affected these areas (e.g., contact restrictions, physical distancing, video-mediated communication). It thus seemed particularly important to investigate the consequences of these changes for mental well-being and quality of life in this group.”
Point 2: In this theoretical framework the authors provide the flowchart typical of statements such as PRISMA. The inclusion and exclusion criteria are not clearly stated. The authors refer to Figure 1 to intuit such criteria
Response 2: As suggested elsewhere in the review, we have now removed the flowchart and included the inclusion and exclusion details in the text: “To identify those publications most relevant to our research question among the massive number of publications in the wake of the COVID-19 pandemic [11], we used Web of Science as our search engine of choice, and began in May 2020 to search for academic publications using the search terms “((Covid OR Corona) AND (ASD OR autis*))”, for the publication years “2020 OR 2021”. This ongoing review of the literature has continued until now, but here we present only papers published up until June 21, 2021. We initially identified 226 papers, of which 84 were excluded because only one of the search terms was relevant (n = 80), because they were duplicated (n = 1), or because we were not able to access them (n = 3). Each of the 142 remaining papers was categorized into one or more of the five topic clusters we identify below, all of which were found to be covered by several papers. Additionally, we distinguished three types of paper: quantitative studies (n = 63), qualitative studies including case reports (n = 20) and opinion papers, topical reviews or editorials (n = 59). Note that we adopted a rather liberal criterion for classifying a study as “quantitative”, whenever the study methods were described in some detail, even if only descriptive data (e.g., percentages) were reported.”
Point 3: They have administered a questionnaire to parents to assess how the pandemic has affected their autistic children, but there is a lack of data on how the contact with those parents and the process developed.
Response 3: We have expanded and highlighted this information by creating a separate “Participants” subsection (2.2) and adding the following information: “Parents were recruited to participate through professional networks, word of mouth, adverts in research and support institutions, and postings in social media.”
Point 4: The main problem of this work is the absence of a common thread that allows the understanding of the work. I recommend the authors to suppress the flowchart and change the structure of the paper.
Response 4: On rereading the article, we had to agree with Reviewer 1 regarding the absence of a common thread. As already mentioned, we have now removed the flowchart and tried to explain the structure of the paper at several points. In addition, and also in response to Reviewer 2, we have moved some text blocks around, and added some new subsections and titles. Our impression is that this presents a considerable improvement.
Reviewer 2 Report
This is a very well-researched paper that includes multiple research elements: lit review, quantitative data (survey), and qualitative data (case reports). I do have a few suggestions to improve the paper:
1) I think the authors are trying to tackle too much. I do not think that the case reports provide anything extra to the paper, and would thus recommend removing them.
2) The Discussion and Conclusions sections include multiple additional references not present in the review portion. This makes the findings confusing and adds too much complexity. These sections are not designed for new references and information, but fitting the paper's findings in the context of the existing lit review. I recommend either removing the new references from these sections, or including them in the original review and referencing them. Especially, there should be no new information in the Conclusions section.
3) While interesting, I agree with the stated premise that the findings are not generalizable to this broader population given the low response rate and use of multiple countries and cultures. Different areas and cultures of the world approach people with autism and related concerns differently, and thus it is difficult to know how to apply the results. This needs to be more explicitly stated in the Limitations.
4) Overall, the multiple limitations listed are downplayed by the authors. There is no need to do this; they should just stand on their own. Don't try to justify them.
5) Another limitation to mention is if there is any thought as to the response rate. Is there a thought that it could be sample bias or response bias, particularly as it relates to the level of support the individuals required? For example, is it likely/possible that parents of these children were more likely to complete it the more support their child needed?
Author Response
First, we would like to thank both reviewers sincerely for their quick but very supportive, to-the-point and helpful reviews. We feel that the changes initiated in response to their comments have really improved the manuscript substantially by increasing readability and integrating the separate parts of the article in to a much more coherent whole.
Point 1: 1) I think the authors are trying to tackle too much. I do not think that the case reports provide anything extra to the paper, and would thus recommend removing them.
Response 1: As suggested, we have removed the case reports as a distinct section. Instead we have acknowledged this anecdotal local evidence in a single paragraph: “At a local level, we can confirm that this was the case: Here in Jena, Germany, the Early Support and Counseling Center was forced by local authorities to stop offering physical special needs and support services in kindergartens, schools, and homes of affected families. The change was sudden, and many parents of children with autism reported that it entailed considerable stress and anxiety due to the loss of daily routines and insights from a third party. It was reported that the younger the affected individuals, the more they struggled with the COVID-19-induced changes. Interestingly, family relationships were generally described as positively different, as more leisure time was spent together than at other times. On a positive note, the Early Support and Counseling Center Jena reported that many adults with autism experienced the lockdown as less problematic and sometimes even as relaxing, due to the reduction of face-to-face social contacts. Many clients also reported positive effects on their family life, which they attributed to more time than before the pandemic spent on activities such as hiking, play, or music – again, an observation that was corroborated by our online survey.”
Point 2: 2) The Discussion and Conclusions sections include multiple additional references not present in the review portion. This makes the findings confusing and adds too much complexity. These sections are not designed for new references and information, but fitting the paper's findings in the context of the existing lit review. I recommend either removing the new references from these sections, or including them in the original review and referencing them. Especially, there should be no new information in the Conclusions section.
Response 2: We have revised the General Discussion substantially by moving the paragraphs concerning further literature to the literature review (“1.8 Other topics of interest”), and agree that they fit here much better. We have removed the “Conclusions” section completely, or rather reframed it as “3.1 Outlook: The importance of social interaction in the face of the pandemic”, because it is important to us to frame the findings in the context of social neuroscience. We hope that Reviewer 2 will agree that first citations here of specific papers are justified, since they go beyond the scope of the literature review.
Point 3: 3) While interesting, I agree with the stated premise that the findings are not generalizable to this broader population given the low response rate and use of multiple countries and cultures. Different areas and cultures of the world approach people with autism and related concerns differently, and thus it is difficult to know how to apply the results. This needs to be more explicitly stated in the Limitations.
Response 3: We share Reviewer 2’s opinion that the generalisation to different countries and cultures is very limited and agree that this was rather implicit in the previous version. We have now added a useful reference and a paragraph on this issue right at the start of the manuscript: “Last but not least, due to the world-wide nature of the pandemic, we were particularly interested to record research findings and survey responses from different countries, cultures, and language backgrounds. This is of interest both because the pandemic and the measures implemented to counter it differed between different countries, but also because societies differ widely in how they view and support autistic children and their families [5]. Of course, a comprehensive comparison of pandemic effects world-wide is well beyond the scope of this article, but we try to maintain a multinational perspective whenever possible.”
In addition, we expanded on the generalisability in the Limitations section of the General Discussion (see Point 5).
Point 4: 4) Overall, the multiple limitations listed are downplayed by the authors. There is no need to do this; they should just stand on their own. Don't try to justify them.
Response 4: We were not entirely sure which part of the text this comment referred to. We hope we may have addressed it in responding to Points 3 and 5, but please let us know if this is not the case.
Point 5: 5) Another limitation to mention is if there is any thought as to the response rate. Is there a thought that it could be sample bias or response bias, particularly as it relates to the level of support the individuals required? For example, is it likely/possible that parents of these children were more likely to complete it the more support their child needed?
Response 5: As already mentioned expanded on the generalisability in the Limitations section of the General Discussion. In particular, we included a discussion of the low response rate and potential response biases: “Regarding our parent-survey of school-age children and adolescents with and without a clinical diagnosis of autism, we fully acknowledge the limitations of our analyses due to the comparatively small number of respondents and their diverse geographical, cultural and economic background. In retrospect, the survey was very long for busy parents to fill out online at a challenging time. It is interesting to speculate about potential response biases here: On the one hand, parents of severely affected children might be particularly likely to fill out the survey in order to vent their frustration, on the other hand, this group is likely to have especially little time to spare for voluntary add-ons such as a survey. Another issue potentially adding to the low response rate is the fact that data collection fell within the summer break in many countries, when daily routines are often suspended even in non-pandemic times. Finally, it goes without saying that the sampled populations in the German-speaking countries, Iran and Turkey cannot not in any way represent a global view. Among many other aspects, they differ both in respect to how ausitm is treated at a societal level [5].”
This manuscript is a resubmission of an earlier submission. The following is a list of the peer review reports and author responses from that submission.